# CPEB2 Suppresses Hepatocellular Carcinoma Epithelial–Mesenchymal Transition and Metastasis through Regulating the HIF-1α/miR-210-3p/CPEB2 Axis

**DOI:** 10.3390/pharmaceutics15071887

**Published:** 2023-07-05

**Authors:** Ran You, Yanjun Yang, Guowen Yin, Hao Jiang, Yousheng Lu, Liang Gui, Jun Bao, Qingyu Xu, Liang Feng

**Affiliations:** 1Department of Interventional Radiology, Jiangsu Cancer Hospital, Jiangsu Institute of Cancer Research, The Affiliated Cancer Hospital of Nanjing Medical University, Nanjing 210009, China; youran@njmu.edu.cn (R.Y.); jsnjygw@njmu.edu.cn (G.Y.); jsnjjh@njmu.edu.cn (H.J.); 2School of Traditional Chinese Pharmacy, China Pharmaceutical University, Nanjing 211198, China; 3121020133@stu.cpu.edu.cn; 3Department of Hepatobiliary Surgery, Jiangsu Cancer Hospital, Jiangsu Institute of Cancer Research, The Affiliated Cancer Hospital of Nanjing Medical University, Nanjing 210009, China; ww_doc@njmu.edu.cn (Y.L.); njmugl@njmu.edu.cn (L.G.); 4Department of Medical Oncology, Jiangsu Cancer Hospital, Jiangsu Institute of Cancer Research, The Affiliated Cancer Hospital of Nanjing Medical University, Nanjing 210009, China; baojun@njmu.edu.cn

**Keywords:** pharmacogenomics, transcriptomics, multi-omics, hepatocellular carcinoma, cytoplasmic polyadenylation element binding protein 2, miR-210-3p

## Abstract

Hepatocellular carcinoma (HCC) is a prevalent and high-mortality cancer worldwide, and its complexity necessitates novel strategies for drug selection and design. Current approaches primarily focus on reducing gene expression, while promoting gene overexpression remains a challenge. In this work, we studied the effect of cytoplasmic polyadenylation element binding protein 2 (CPEB2) in HCC by constructing tissue microarrays (TAMs) from 90 HCC cases and corresponding para-cancerous tissues. Our analysis showed that CPEB2 expression was significantly reduced in HCC tissues, and its low expression was associated with a higher recurrence risk and poorer prognosis in patients with head and neck cancer. CPEB2 was found to regulate HCC epithelial–mesenchymal transition (EMT) and metastasis through the HIF-1α/miR-210-3p/CPEB2 feedback circuit. Using the RNA binding protein immunoprecipitation (RIP) assay, we demonstrated that miR-210 directly governs the expression of CPEB2. The inverse relationship between CPEB2 expression and miR-210-3p in HCC tissues suggested that this regulatory mechanism is directly linked to HCC metastasis, EMT, and clinical outcomes. Moreover, utilizing the SM2miR database, we identified drugs that can decrease miR-210-3p expression, consequently increasing CPEB2 expression and providing new insights for drug development. In conclusion, our findings illustrated a novel HIF-1α/miR-210-3p/CPEB2 regulatory signaling pathway in HCC and highlighted the potential of enhancing CPEB2 expression through targeting miR-210-3p as a novel predictive biomarker and therapeutic strategy in HCC, as it is modulated by the HIF-1α/miR-210-3p/CPEB2 feedback circuit.

## 1. Introduction

Hepatocellular carcinoma (HCC), with roughly 900,000 new cases and 830,000 fatalities per year, is the sixth most frequent malignancy and the third major cause of cancer-related deaths globally [1]. HCC is expected to be the third most common cancer-related cause of fatalities in America by 2030. The increased mortality rate linked to HCC is attributed mostly to its aggressive character, marked by high recurrence and metastatic rates. Therefore, understanding the mechanisms governing HCC invasiveness and metastasis is of paramount importance to develop novel therapeutic strategies and targets.

The cytoplasmic polyadenylation element binding (CPEB) protein family (CPEB1-4) is essential for gene expression regulation after transcription. These proteins have a common C-terminal RNA-binding domain responsible for binding to specific cytoplasmic polyadenylation elements inside target mRNAs’ 3’ untranslated region, facilitating translation. The CPEB family plays a significant role in maintaining cellular homeostasis in somatic tissues and regulates various processes, such as synaptic plasticity, cell proliferation, polarity, and senescence [2]. CPEB2 belongs to the CPEB protein family, which mediates cytoplasmic polyadenylation of target mRNAs and regulates their translation efficiency [3,4,5,6]. However, the mechanism that influences CPEB2 in HCC is not comprehensively known and needs to be further explored. Epithelial–mesenchymal transition (EMT) plays a key role in driving tumor cell metastasis by modulating cell–cell contact, cell–extracellular matrix (ECM) interactions, and orchestrating the phenotypic transfer of an epithelial cell to a mesenchymal state [7,8,9]. The complex EMT regulatory network involves the interplay of long non-coding RNAs, microRNAs (miRNAs), transcription factors, and other signaling molecules [10,11,12]. Accumulating evidence suggests that miRNAs, particularly miR-210-3p, play vital roles in the development, progression, and metastasis of HCC [13,14]. MiR-210-3p as an oncogene is a hypoxia-responsive miRNA that promotes angiogenesis, metastasis, and tumor growth in various cancers [15,16,17,18,19]. In addition, hypoxia-inducible factor 1α (HIF-1α) has been linked to mediating adaptive responses to hypoxia and regulating EMT and metastasis in HCC as a transcription factor [20,21]. However, the precise functional action of the HIF-1α/miR-210-3p axis within the HCC EMT remains to be elucidated.

The development of pharmacological interventions to increase gene expression, especially that of tumor suppressor genes, has proven to be challenging. By integrating transcriptomics and pharmacogenomics approaches, a novel strategy for drug design and selection can be devised. Our study aims to increase CPEB2 expression by targeting miR-210-3p, thus providing a new avenue for HCC drug discovery. In this investigation, we found a strong correlation between reduced CPEB2 expression and poor clinical outcomes in HCC patients. Functionally, CPEB2 exhibits tumor suppressor properties in HCC by inhibiting cell motility and colony formation in vitro and attenuating metastasis in vivo. Furthermore, CPEB2 expression governs EMT phenotypes, preventing HCC cell metastasis. We also elucidated the regulatory mechanism of miR-210-3p, a downstream target of CPEB2 in EMT and HCC metastasis. The interaction of miR-210-3p with CPEB2 may serve as a valuable prognostic indicator for HCC, paving the way for a new strategy for drug design and selection by combining transcriptomics and pharmacogenomics approaches.

## 2. Methods

### 2.1. Transcriptome Analysis

We performed microarray analysis to investigate the gene expression profiles of our samples. Isolation of total RNA from the cells was carried out following the manufacturer’s instructions for the tissue using TRIzol reagent (Invitrogen, Waltham, MA, USA). A NanoDrop spectrophotometer (Thermo Fisher Scientific, life nanodrop2000, Waltham, MA, USA) and agarose gel electrophoresis were employed to analyze the purity and concentration of the RNA. The extracted RNA was then reverse transcribed into cDNA, labeled with fluorescent dyes (Cy3 and Cy5), and hybridized to a whole-genome microarray chip (Agilent Technologies, SurePrint G3, Santa Clara, CA, USA) following the manufacturer’s guidelines. After hybridization, cleaning and scanning of the microarray slides with an Agilent microarray scanner (Agilent Technologies, G4900DA, Santa Clara, CA, USA) were performed to obtain fluorescence intensity data for each spot on the array. Agilent feature extraction software 12.2.0.7 version (Agilent Technologies) was applied for processing the raw data, and the quantile normalization approach was utilized to standardize the data. A fold change threshold of ≥2 and an adjusted *p*-value of ≤0.05 were utilized to identify differentially expressed genes.

### 2.2. Clinical Samples and Cell Lines

Cell lines for hepatocellular carcinoma (HCC) (MHCC-97L, SMMC-7721, HepG2, and Huh-7) were acquired from the Chinese Academy of Sciences cell bank. Tumor samples and matched non-tumor tissues were taken from 90 HCC patients who underwent surgical resection at Jiangsu Cancer Hospital from 2015–2017. The cohort comprised 80 men (88.89%) and 10 women (11.11%), and the average age was 52.3 years (range: 31–78 years). In total, 71 of the 90 patients (78.9%) had chronic hepatitis B virus infection. The patients had not received any systemic or local therapy before enrollment. Surgical specimens were either stored at −80 °C or fixed in 10% formalin buffer for paraffin embedding. This study involving human tissue was approved by the ethical review committee of Nanjing Medical University (grant number: 2019050) and informed written consent was obtained from the patients in accordance with the Declaration of Helsinki. All of the studies involving human participants were blinded. In Table 1, the demographic characteristics of the HCC patients are shown.

### 2.3. Immunohistochemistry

Immunohistochemical detection of CPEB2, E-cadherin, and vimentin expression was performed on 10 μm thick, paraffin-embedded tissue sections. Primary antibodies against CPEB2 (1:100 dilution), E-cadherin, and vimentin (1:150 dilution, Cell Signaling Technology, Danvers, MA, USA) were applied. Two independent pathologists analyzed the results. The CPEB2 expression score was determined based on the percentage of CPEB2-positive cells (<10% positive cells = 0, 10–25% = 1, 26–50% = 2, 51–75% = 3, and 76–100% = 4). Low CPEB2 expression groups had scores of 0, 1, or 2, while high expression groups had scores of 3 or 4.

### 2.4. Western Blot

The cells were harvested on ice in lysis buffer with a protease inhibitor cocktail, and cell debris were removed using 5 min centrifugation at 13,000× *g*. A BCA protein assay kit was used to determine the total protein concentration, and the lysates were separated by SDS-PAGE. The nitrocellulose membranes were closed with 5% skim milk in TBST and overnight incubation with a primary antibody at 4 °C: CPEB2, HIF-1α, anti-E-cadherin (1:1000), anti-vimentin (1:1000), anti-GAPDH (1:500), and IgG (1:3500). On the second day, secondary antibodies and ECL detecting agents were used according to standard procedures.

### 2.5. Cell Migration and Invasion Assays

Transwell membranes (Corning, Corning, NY, USA) were used for cell migration and invasion tests following the manufacturer’s instructions. For 30 min, the cells were fixed and stained in methanol with crystal violet (Sigma, St. Louis, MO, USA). The transfected cells were planted onto 6-well plates for the wound healing experiment, and a cell monolayer was scraped with a 100 L pipette tip. Images were obtained at zero and twenty-four hours following injury.

### 2.6. Luciferase Reporter Assay

To build a luciferase reporter test, a wild-type CPEB2 segment was introduced into the Renilla luciferase gene (Promega, Madison, WI, USA). The mutant segment of CPEB2 was produced by inserting the mutated sequences that could bind the sites of miR-210-3p, named CPEB2 MUT. For the HIF-1α luciferase activity assay, the cells were cultured under normoxic or hypoxic conditions for 24 h and transfected with the luciferase reporter vectors. The relative luciferase activity was determined after normalizing the firefly luciferase activity to the reninase activity.

### 2.7. RNA-Immunoprecipitation (RIP) Assay

The cells were lysed in RIP lysis buffer according to the manufacturer’s instructions, and the lysates were treated with magnetic beads conjugated to particular antibodies or control IgG. After incubation, bead-bound RNA–protein complexes were washed, and co-precipitated RNAs were eluted and purified. The isolated RNAs were then subjected to quantitative real-time PCR analysis to inspect the specific RNAs of interest.

### 2.8. Cell Migration and Invasion Assays

A modified Boyden chamber experiment with Transwell inserts was employed to measure cell migration and invasion capabilities. In a serum-free medium, the cells were seeded into the top chamber of the inserts for migration tests. The top chamber was pre-coated with Matrigel for the invasion experiments. In both cases, full medium including serum was employed as a chemoattractant. Non-migrating or non-invading cells on the membrane’s top side were carefully removed with a cotton swab after incubation. Under a microscope, fixed cells that had passed the membrane were stained with crystal violet and counted.

### 2.9. Quantitative Real-Time PCR (qPCR)

Total RNA was extracted from the cells as directed by the manufacturer using an RNA isolation kit. A NanoDrop spectrophotometer was used to determine RNA concentration and quality. A reverse transcription kit with oligo(dT) primers was used to create complementary DNA (cDNA). In a real-time PCR system, qPCR was performed with an SYBR-Green-based master mix and gene-specific primers. The housekeeping gene GAPDH was used to normalize target gene expression, and relative expression levels were calculated using the 2^−ΔΔCt^ method.

### 2.10. Animal Experiments

The animal tests were authorized by the Animal Ethics Committee at Nanjing Medical University (Nanjing, China). Four-week-old male BALB/C nude mice were obtained from Nanjing Medical University’s Experimental Animal Center and acclimated for one week before the study began. Under 12:12 light–dark settings, the mice were provided free access to food and drink. The HCC cell lines (5 × 10^6^) were delivered intravenously into the caudal vein to induce lung metastasis. The mice were euthanized after 2 months, and lung metastasis was confirmed by hematoxylin–eosin (H and E) staining. In a separate experiment, tumor growth was assessed every three days after subcutaneous injection of the nude mice with stably produced shNC or shCPEB2 cells. Six weeks after injection, the mice were euthanized, and their tumor weight was measured.

### 2.11. Statistical Analysis

Associations between CPEB2 and miR-210-3p expression levels, as well as the chi-square test, were used to examine the patients’ clinicopathological features. Depending on the data distribution, parametric (two-sample *t*-test) and non-parametric (Mann–Whitney U test) tests were used for continuous data. Kaplan–Meier survival curves were created to examine the survival data, and variations in survival rates were evaluated using the log-rank test. Statistical significance was determined using appropriate tests.

### 2.12. Pharmacological Modulation of miRNA Expression

To identify small molecules capable of modulating miRNA expression using the SM2miR library, a high-throughput screening method was applied. The library consisted of a collection of bioactive small molecules with known targets and mechanisms of action. In 96-well plates, cells were sown and left to adhere overnight. The next day, the cells were treated with individual compounds from the SM2miR library at a predetermined concentration. After 48 h of incubation, the cells’ total RNA was isolated using a commercially available RNA isolation kit following the manufacturer’s instructions. Subsequently, the expression levels of miR-210-3p and other relevant miRNAs were quantified using quantitative real-time PCR (qPCR) as described earlier. Small molecules that significantly altered miRNA expression were identified by comparing the treated cells to the vehicle control. Hits were further validated using dose–response experiments to assess the potency and efficacy of the compounds. Selected small molecules were then evaluated for their effects on the downstream targets and biological processes associated with the miRNA of interest, including cell migration, invasion, and proliferation assays.

## 3. Results

### 3.1. Transcriptomics Analysis

Three pairs of HCC tumor (T) and non-tumor (NT) tissues were analyzed for gene expression. Among the CPEB family, only CPEB2 exhibited downregulation in HCC (Figure 1A,B). Immunohistochemical analysis revealed a significant decrease in CPEB2 expression in T compared to NT tissues (Figure 1C). High CPEB2 expression was observed in 18.9% (17/90) of the T samples, while 90% (81/90) of the NT samples exhibited high CPEB2 expression (*p* < 0.001, Figure 1D,E).

### 3.2. Upregulation of CPEB2 Expression Reduces the Migration and Invasion of HCC Cell Lines

The highest and lowest expression of CPEB2 in the HCC cell lines was for HepG2 and SMMC-7721, respectively (Figure 2A). CPEB2 siRNA or using the overexpression vector confirms successful downregulation or upregulation of CPEB2 (Figure 2B,C). The wound healing, migration, and invasion assays demonstrated that downregulation of CPEB2 may increase HepG2 cell motility and invasion (Figure 2D,E). Similarly, the upregulation of CPEB2 decreased migration and invasion in SMMC-7721 cells (Figure 2F,G).

### 3.3. Lung Metastasis and EMT Phenotypes in Xenograft Models

The effect of CPEB2 on lung metastasis was investigated using a tail vein injection nude mouse model. Mice injected with SMMC-7721-shCPEB2 cells displayed increased lung metastasis compared to the SMMC-7721-shNC group (Figure 3A,C). Additionally, the downregulation of CPEB2 in the MHCC-97L cells led to an increased number of metastatic foci in lung tissues (Figure 3B,D). The influence of CPEB2 on EMT phenotype expression was then examined in the subcutaneous xenograft model. Although no discernible difference in tumor size was observed across the groups (Figure 3E,F), the shCPEB2 group for SMMC-7721 and MHCC-97L cells had considerably higher E-cadherin levels and lower vimentin levels (Figure 3G). 

### 3.4. CPEB2 Impedes EMT Processes in HCC

The immunoblotting data demonstrated downregulation of CPEB2 in the HepG2 cells, leading to increased expression of e-calmodulin and vimentin, whereas upregulation of CPEB2 in SMMC-7721 cells produced the opposite effect on EMT markers (*p* < 0.01, Figure 4A,B). Immunohistochemical analysis of EMT markers in HCC tissues revealed that patients with high CPEB2 expression exhibited reduced E-cadherin and increased vimentin levels, while those with low CPEB2 expression showed an inverse relationship (Figure 4C).

### 3.5. CPEB2 Is the Target of miR-210-3p

To investigate the potential molecular mechanisms by which CPEB2 modulates HCC metastasis, we employed a public database (Targetscan) to identify candidate genes directly targeted by CPEB2. miR-210-3p was found to directly bind to the 3’UTR of CPEB2 (Figure 5A), suggesting a functional relationship. In the HepG2 cells, miR-210-3p downregulation increased luciferase activity in the Wt-CPEB2 3’UTR but not in the Mt-CPEB2 3’UTR (Figure 5B). Similarly, in the SMMC-7721 cells, upregulation of miR-210-3p decreased luciferase activity in the Wt-CPEB2 3’UTR, while the Mt-CPEB2 3’UTR remained unaffected (Figure 5C). The results of this study showed that miR-210-3p binds to the 3’UTR of CPEB2, which is the target of miR-210-3p. Immunoblotting and RT-PCR analyses confirmed that miR-210-3p modulates CPEB2 expression in HCC cell lines (Figure 5D–G). A negative connection between CPEB2 and miR-210-3p was found in the HCC tumor tissues (Figure 5H and Table 2).

### 3.6. CPEB2 Is Crucial in miR-210-3p‘s Influence on HCC Metastasis and EMT Processes

Based on the above findings, we looked into the involvement of CPEB2 in miR-210-3p’s biological activity in HCC metastasis further. In the anti-miR-210-3p treatment of the HepG2 cells, CPEB2 siRNA effectively reduced CPEB2 expression, thereby enhancing miR-210-3p-induced EMT processes, as evidenced by decreased E-cadherin and increased vimentin levels (Figure 6A). The results revealed that CPEB2 downregulation in anti-miR-210-3p-treated HepG2 cells promoted HCC migration and invasion in the cell migration and invasion assays (Figure 6C,D). In contrast, CPEB2 expression was upregulated in the SMMC-7721 cells treated with miR-210-3p mimics, leading to the opposite influence on HCC invasion and migration (Figure 6B,E,F). The above results revealed that miR-210-3p’s influence on HCC metastasis and EMT processes is mediated through CPEB2.

### 3.7. Hypoxia Modulates CPEB2 via HIF-1α/miR-210-3p/CPEB2 Regulatory Circuit

Under hypoxic conditions, miR-210-3p expression increased (Figure 7A) and HIF-1α luciferase activity was enhanced, while the inhibition of miR-210-3p decreased the activity of HIF-1α (Figure 7B). HIF-1α inhibition induced CPEB2 suppression (Figure 7C,D). As depicted in Figure 7E, CPEB2 protein expression was downregulated under hypoxia but restored by anti-miR-210-3p treatment. Concomitantly, miR-210-3p expression significantly decreased when CPEB2 was upregulated under hypoxic conditions (Figure 7F). To investigate whether CPEB2 modulates HIF-1α expression, dual-luciferase reporter assays were performed. Upregulating CPEB2 reduced luciferase activity under hypoxia, but no significant differences were observed under normoxic conditions or when co-transfected with the mutant type (Figure 7G). CPEB2 upregulation inhibited HIF-1α expression, while its downregulation produced the opposite effect (Figure 7H). The RIP experiments revealed that CPEB2 could bind to HIF-1α mRNA (Figure 7I), suggesting that CPEB2 is regulated via a HIF-1α/miR-210-3p/CPEB2 feedback loop.

### 3.8. Clinical Relevance of CPEB2 and miR-210-3p Expression in HCC Patients

A significant association was identified between reduced CPEB2 levels and increased recurrence risk (*p* = 0.011, Table 1), as well as a positive relationship, which exists between miR-210-3p levels and vascular invasion and recurrence (*p* = 0.034 and *p* = 0.044, respectively, Table 1). The potential prognostic roles of both CPEB2 and miR-210-3p in HCC were assessed. Patients with high CPEB2 expression exhibited longer overall survival (OS) (*p* = 0.038, Figure 8A) and progression-free survival (PFS) (*p* = 0.024, Figure 8B), while low miR-210-3p expression was related to better OS (*p* = 0.001) and PFS (*p* = 0.017, Figure 8C,D). According to the combined survival prognosis based on CPEB2 and miR-210-3p expression, patients with high miR-210-3p and low CPEB2 levels had the shortest OS and PFS. Conversely, patients with high CPEB2 and low miR-210-3p expression experienced the longest OS and PFS (*p* = 0.002 and *p* = 0.004, respectively, Figure 8E,F). These findings indicate that a combination of CPEB2 and miR-210-3p may serve as a prognostic indicator for HCC diagnosis and treatment.

### 3.9. Potential Therapeutic Targets for miR-210 Modulation

Based on the information from the SM2miR database, several small molecules were identified to downregulate miR-210 expression. These small molecules include lenalidomide, arsenic trioxide, 5-Fluorouracil, ginsenoside Rh2, Aidi injection, 5-aza-2′-deoxycytidine, and trastuzumab. These potential therapeutic targets could play a role in modulating the miR-210/CPEB2 interaction; as a result, innovative techniques for the diagnosis and treatment of HCC are now available (Table 3).

## 4. Discussion

HCC is the most common primary liver cancer and the third leading cause of cancer-related death worldwide. In recent years, chemical agents, such as sorafenib, envatinib, and regorafenib, have been approved for the treatment of advanced HCC [22]. However, there are deficiencies in their clinical efficacy, toxicity, and the need for clinically applicable and validated predictive biomarkers [23]. Currently, there are still limited clinically applicable and validated predictive biomarkers to identify HCC patients who would benefit from systemic therapy. Further prospective biomarker validation studies of personalized systemic therapy for HCC are needed.

In the research, we uncovered CPEB2’s important involvement in inhibiting HCC metastasis and EMT phenotypes by regulating the miR-210-3p pathway. Our findings elucidate a unique regulatory mechanism involving the HIF-1α/miR-210-3p/CPEB2 feedback loop under hypoxic conditions and suggest potential therapeutic targets for HCC. In the HCC tissues, our data demonstrated that CPEB2 was significantly downregulated, and its lack of expression was linked to poor clinical outcomes, including tumor recurrence and vascular invasion, which is consistent with the findings already reported in previous studies [24]. This observation highlights, in HCC progression, the tumor-suppressive role of CPEB2. Moreover, we found that CPEB2 inhibited EMT phenotypes in HCC, further supporting its role as a crucial factor in suppressing HCC metastasis, which has also been previously reported to promote differentiation and suppress EMT in a variety of cancer cells [25].

One of the novel findings in the research is the identification of miR-210-3p as a direct downstream target of CPEB2. We showcase that miR-210-3p could regulate CPEB2 expression, and they were inversely correlated in HCC tumor tissues. Furthermore, we proved that miR-210-3p played a significant role in HCC metastasis and the EMT process, which was mediated by CPEB2. We also investigated the effects of hypoxia in regulating the HIF-1α/miR-210-3p/CPEB2 feedback loop. A related study evaluated the changes in HIF-1α mRNA levels after altering miR-210 expression, and the HIF-1α levels were also negatively correlated with the miR-210 levels, a finding that is consistent with the results found in our study [26,27,28]. Our results showed that hypoxia increased the expression of miR-210-3p, while the inhibition of miR-210-3p decreased HIF-1α activity. Additionally, we found that CPEB2 could regulate HIF-1α expression, indicating a complex regulatory network involving hypoxia, HIF-1α, miR-210-3p, and CPEB2. At the same time, miR-210 has been identified as a standalone prognostic factor for HCC, positively correlating with its angiogenesis [29,30]. Our analysis of the SM2miR database revealed several small molecules that could potentially modulate miR-210 expression, such as lenalidomide, arsenic trioxide, 5-Fluorouracil, ginsenoside Rh2, Aidi injection, 5-aza-2’-deoxycytidine, and trastuzumab. These molecules could be explored as novel therapeutic targets for HCC, acting through the miR-210-3p/CPEB2 interaction. According to research reports on the analysis of miRNA–drug relationships, arsenic trioxide could inhibit the progression of leukemia by upregulating the expression of some miRNAs, including miR-150-5p and miR-96-5p [31]. Similarly, 5-fluorouracil can significantly dysregulate the expression level of miR-1246 in cancer cells [32]. All of the above reports indicate that these small molecules can upregulate or downregulate the expression levels of specific miRNAs and can be used as adjuncts as potential supplements to standard clinical treatment for cancer. Furthermore, follow-up studies combined with the analysis of the SM2miR database may validate the predicted interactions between small molecule compounds and miR-210.

Finally, our findings emphasize the critical function of CPEB2 in decreasing HCC metastasis by inhibiting EMT phenotypes via the miR-210-3p pathway. The involvement of hypoxia and the HIF-1α/miR-210-3p/CPEB2 regulatory loop adds another layer of complexity to the molecular mechanisms underlying HCC progression. The identification of small molecules that modulate miR-210 expression offers potential therapeutic targets for HCC treatment. This requires further research to verify the clinical relevance of these findings and to develop effective therapeutic strategies targeting the miR-210-3p/CPEB2 interaction in HCC. However, there are still some limitations to this study, and further experiments in vivo should be performed to explore the role of CPEB2 in the metastasis of hepatocellular carcinoma in an animal model. The relationship between the levels of CPEB2 and the prognosis of patients with HCC at different stages should also be analyzed.

## 5. Conclusions

In hepatocellular carcinoma tissues, CPEB2 expression was downregulated and associated with miR-210-3p antagonism. Moreover, our study demonstrated that CPEB2 is downregulated in HCC cell lines and tissues and that low CPEB2 expression is associated with metastasis and the EMT process. This research provides mechanistic evidence that the HIF-1/miR-210-3p/CPEB2 feedback loop plays a regulatory role in CPEB2 expression. Therefore, it is conceivable to believe that this feedback loop represents a potential mechanism linking hypoxia to the regulation of EMT and metastasis in HCC through the regulation of CPEB2 in hypoxic HCC cells and tissues. In general, our findings revealed that CPEB2 may be a future diagnostic marker and precision medicine target for HCC, especially in individualized treatment that takes into account the influence of genetics on efficacy.

## Figures and Tables

**Figure 1 pharmaceutics-15-01887-f001:**
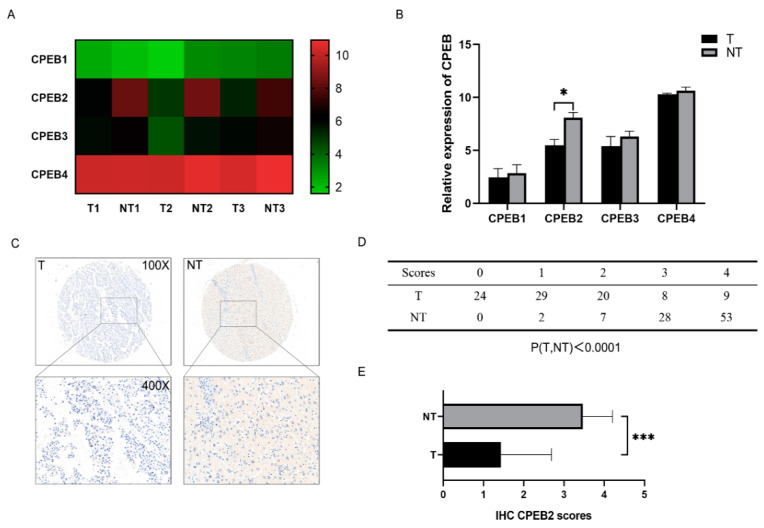
Reduced expression of CPEB2 in HCC. (**A**,**B**) Heat map (**A**) and quantitative analysis of CPEB family expression in three pairs of tumor (T) and corresponding non-tumor (NT) liver tissues in HCC (**B**); (**C**) immunohistochemical staining of CPEB2 in T and NT liver tissues; and (**D**,**E**) statistical analysis of grouped cases according to immunohistochemical scores. T vs. NT, * *p* < 0.05, *** *p* < 0.001.

**Figure 2 pharmaceutics-15-01887-f002:**
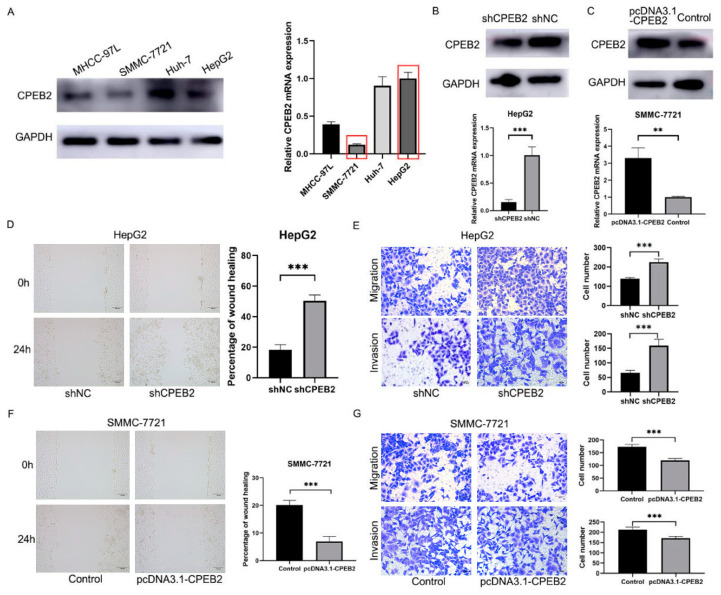
CPEB2 suppresses HCC cell migration and invasion in vitro. (**A**) CPEB2 expression in different HCC cell lines determined by immunoblotting and RT-PCR analysis (red box: lowest and highest expression of CPEB2 in SMMC-7721 and HepG2 cell, respectively); (**B**,**C**) immunoblotting and RT-PCR analysis of CPEB2 in HepG2 cells with downregulated CPEB2 (**B**) and SMMC-7721 cells with upregulated CPEB2 (**C**). (**D**,**E**) Wound healing (**D**), migration, and invasion assays (**E**) of HepG2 cells with shNC or shCPEB2. (**F**,**G**) Wound healing (**F**), migration, and invasion assays (**G**) of SMMC-7721 cells with the control or pcDNA3.1-CPEB2. shCPEB2 vs. shNC, pcDNA3.1-CPEB2 vs. the control, *** *p* < 0.001; pcDNA3.1-CPEB2 vs. the control, ** *p* < 0.01. The red boxes represent the lowest and highest expression of CPEB2 in SMMC-7721 and HepG2 cell respectively, respectively. Scale bar: 100 μm (**D**,**F**); 50 μm (**E**,**G**).

**Figure 3 pharmaceutics-15-01887-f003:**
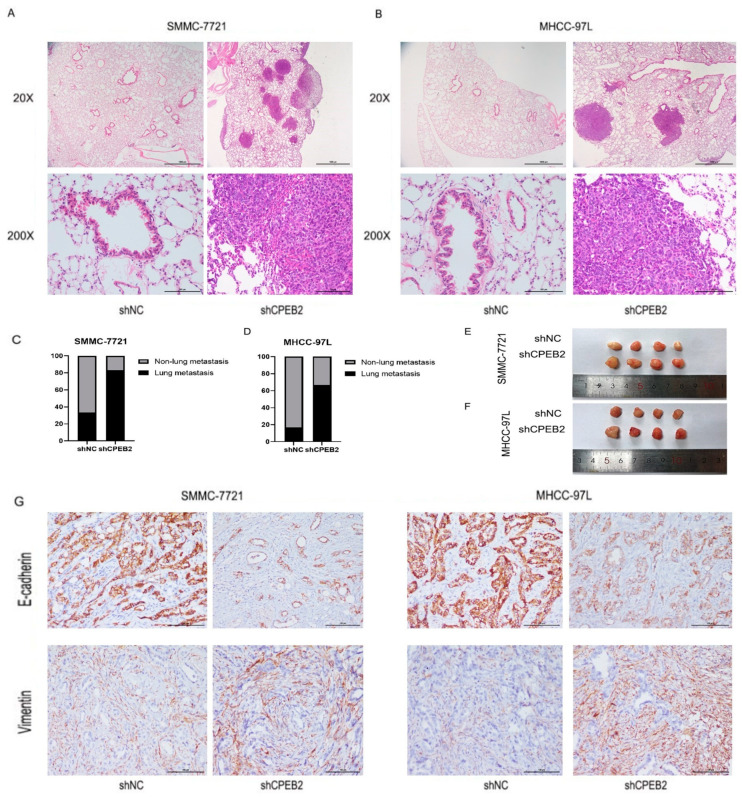
CPEB2 inhibited lung metastasis of HCC and EMT phenotypes in the xenograft models. (**A**,**B**) Representative H and E staining of lung metastasis formed by SMMC-7721 (**A**) and MHCC-97L (**B**) cells. (**C**,**D**) Percentage of mice with lung metastasis formed by SMMC-7721 (**C**) and MHCC-97L (**D**) cells. (**E**,**F**) Representative pictures of xenograft tumors caused by subcutaneously injecting SMMC-7721 (**E**) or MHCC-97L (**F**) cells. (**G**) E-cadherin and vimentin expression immunohistochemically stained in the subcutaneous xenograft model (scale bar: 100 μm).

**Figure 4 pharmaceutics-15-01887-f004:**
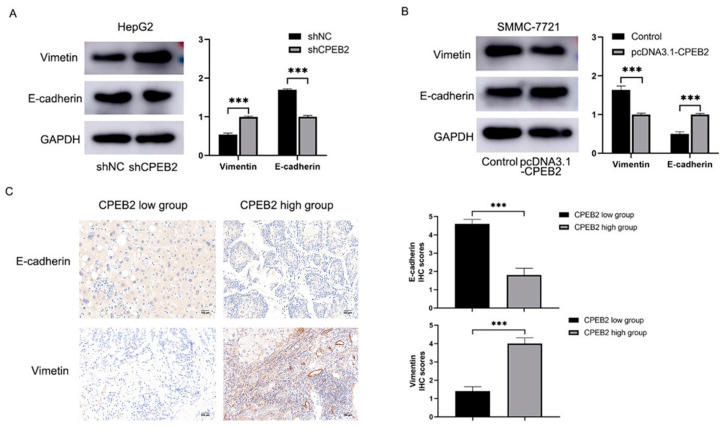
CPEB2 suppresses EMT phenotypes in HCC. (**A**,**B**) Immunoblotting of E-cadherin and vimentin after CPEB2 downregulation in HepG2 cells (**A**) and upregulation in SMMC-7721 cells (**B**). (**C**) Immunohistochemistry of E-cadherin and vimentin in HCC tumor tissues with high or low CPEB2 levels. shCPEB2 vs. shNC, CPEB2 high group vs. CPEB2 low group (scale bar: 100 μm), *** *p* < 0.001.

**Figure 5 pharmaceutics-15-01887-f005:**
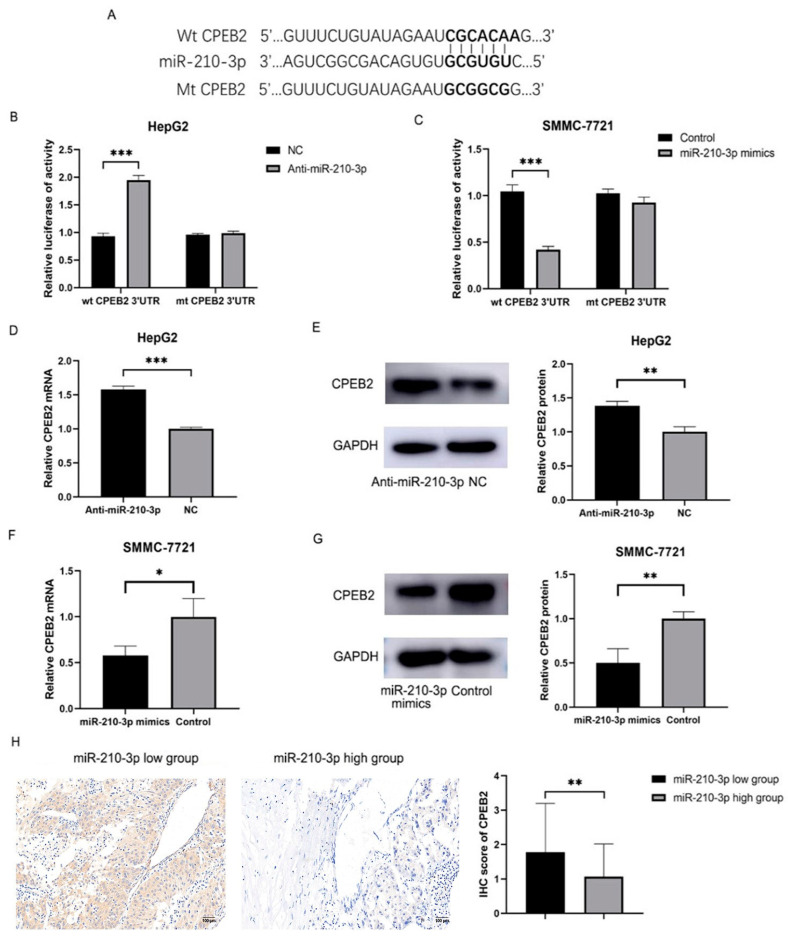
CPEB2 targets and regulates miR-210-3p. (**A**) Wild-type (WT) and mutant variant (MT) of the putative miR-210-3p target sequences within the CPEB2 gene. (**B**,**C**) Luciferase reporter assays of miR-210-3p with WT or MT-CPEB2 in HepG2 (**B**) and SMMC-7721 (**C**) cells. (**D**,**E**) RT-PCR (**D**) and immunoblotting (**E**) analysis of CPEB2 in HepG2 cells after NC or anti-miR-210-3p treatment. (**F**,**G**) RT-PCR (**F**) and immunoblotting (**G**) analysis of CPEB2 in SMMC-7721 cells after control or miR-210-3p mimic treatment. (**H**) Immunohistochemistry of CPEB2 in HCC tumor tissues with high or low miR-210-3p levels (scale bar: 100 μm). shCPEB2 vs. shNC, * *p* < 0.05, ** *p* < 0.01, *** *p* < 0.001; pcDNA3.1-CPEB2 vs. Control, ** *p* < 0.01, *** *p* < 0.001.

**Figure 6 pharmaceutics-15-01887-f006:**
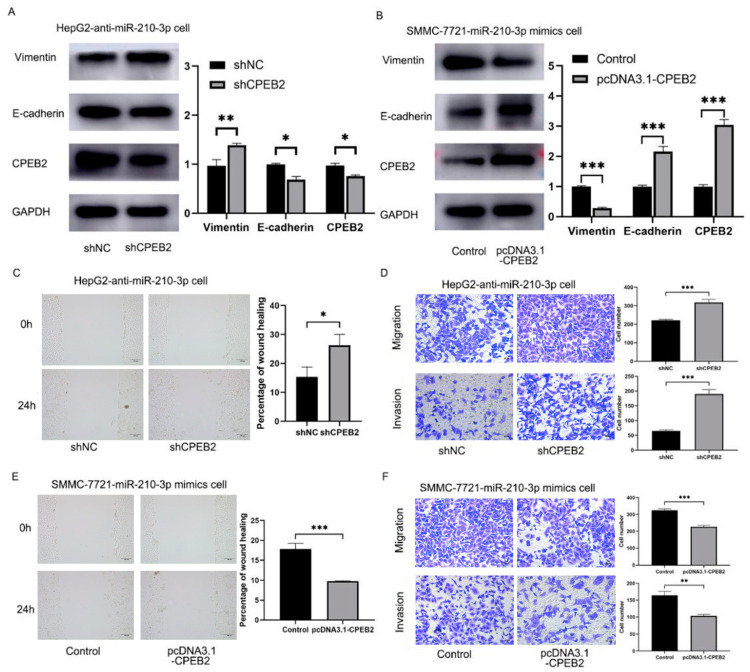
CPEB2 is essential for miR-210-3p’s effects on HCC metastasis and EMT processes. (**A**,**B**) Immunoblotting analysis of CPEB2, E-cadherin, and vimentin in anti-miR-210-3p HepG2 cells treated with shNC or shCPEB2 (**A**) and in miR-210-3p mimic SMMC-7721 cells treated with the control or pcDNA3.1-CPEB2 (**B**). (**C**,**D**) Wound healing (**C**) and migration and invasion assays (**D**) of anti-miR-210-3p HepG2 cells with shNC or shCPEB2. (**E**,**F**) Wound healing (**E**) and migration and invasion assays (**F**) of miR-210-3p mimic SMMC-7721 cells with the control or pcDNA3.1-CPEB2. Anti-miR-210-3p vs. NC, ** *p* < 0.01, *** *p* < 0.001; miR-210-3p mimics vs. the control, * *p* < 0.05, ** *p* < 0.01, *** *p* < 0.001; miR-210-3p high group vs. miR-210-3p low group, ** *p* < 0.01. Scale bar: 100 μm (**C**,**E**); 50 μm (**D**,**F**).

**Figure 7 pharmaceutics-15-01887-f007:**
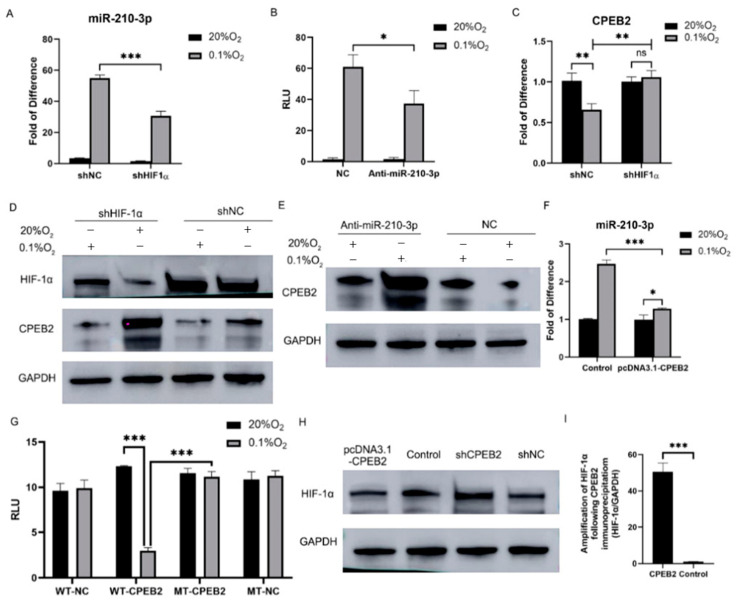
Hypoxia suppresses CPEB2 through the HIF-1α/miR-210-3p/CPEB2 regulatory circuit. (**A**) miR-210-3p levels under normoxic or hypoxic conditions determined by RT-PCR. (**B**) Luciferase reporter assay of HIF-1α activities under normoxic or hypoxic conditions in the NC or anti-miR-210-3p-treated groups. (**C**) mRNA levels of CPEB2 in the shNC and shHIF-1α groups treated with or without hypoxia, determined by RT-PCR (ns: no significance). (**D**) Immunoblotting analysis of HIF-1α and CPEB2 in the shNC and shHIF-1α groups under normoxic or hypoxic conditions. (**E**) Immunoblotting analysis of CPEB2 in the NC and anti-miR-210-3p groups under normoxic or hypoxic conditions. (**F**) miR-210-3p levels in the control and pcDNA3.1-CPEB2 groups under normoxic or hypoxic conditions, determined by RT-PCR. (**G**) Luciferase reporter assay of CPEB2 with WT or MT-CPEB2 under normoxic or hypoxic conditions. (**H**) Immunoblotting analysis of HIF-1α in the shNC, shCPEB2, control, or pcDNA3.1-CPEB2 groups. (**I**) Quantification of HIF-1α mRNA bound to individual FLAG-tagged CPEB2 proteins immunoprecipitated with anti-FLAG antibody. 20% O_2_ vs. 0.1% O_2_, * *p* < 0.05, ** *p* < 0.01, *** *p* < 0.001; CPEB2 vs. the control, *** *p* < 0.001.

**Figure 8 pharmaceutics-15-01887-f008:**
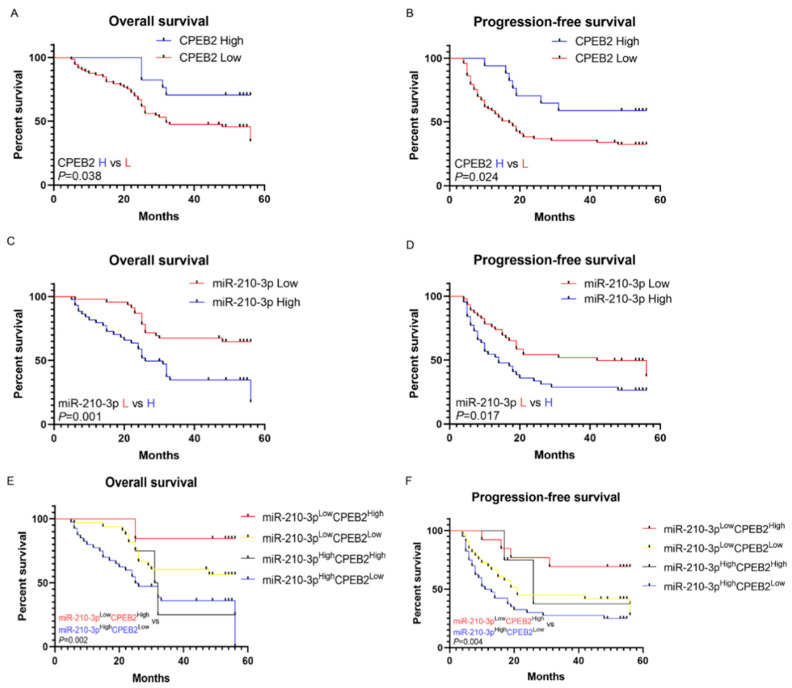
Clinical significance of CPEB2 and miR-210-3p expression in HCC patients. (**A**,**B**) Association of CPEB2 expression with overall survival (OS) (**A**) and progression-free survival (PFS) (**B**). (**C**,**D**) Association of miR-210-3p expression with OS (**C**) and PFS (**D**). (**E**,**F**) Association of the combination of CPEB2 and miR-210-3p expression with OS (**E**) and PFS (**F**).

**Table 1 pharmaceutics-15-01887-t001:** Clinical characteristics of 90 HCC patients analyzed.

Clinicopathological Features	Total No. of Cases *n* = 90	No. of Cases with CPEB2^Low^	No. of Cases with CPEB2^High^	*p*-Value	No. of Cases with miR-210-3p^Low^	No. of Cases with miR-210-3p^High^	*p*-Value
Sex	-	-	-	-	-	-	-
Male	80	64	16	0.739	42	38	0.682
Female	10	9	1	-	4	6	-
Age, year	-	-	-	-	-	-	-
<50	37	29	8	0.580	16	21	0.212
≥50	53	44	9	-	30	23	-
Pathological grades	-	-	-	-	-	-	-
I–II	43	34	9	0.636	20	23	0.404
III–IV	47	39	8	-	26	21	-
Vascular invasion	-	-	-	-	-	-	-
No	43	32	11	0.121	27	16	**0.034** *
Yes	47	41	6	-	19	28	-
Tumor size, cm	-	-	-	-	-	-	-
<5	55	43	12	0.422	29	26	0.701
≥5	35	30	5	-	17	18	-
Recurrence	-	-	-	-	-	-	-
No	34	23	11	**0.011** *	22	12	**0.044** *
Yes	56	50	6	-	24	32	-
Tumor number	-	-	-	-	-	-	-
Single	79	65	14	0.728	42	37	0.296
Multiple	11	8	3	-	4	7	-
Tumor capsular	-	-	-	-		-	-
Complete	43	31	12	**0.037** *	26	17	0.090
Incomplete	47	42	5	-	20	27	-
Liver cirrhosis	-	-	-	-	-	-	-
No	9	7	2	1.000	5	4	1.000
Yes	81	66	15	-	41	40	-
HBsAg	-	-	-	-	-	-	-
Negative	19	14	5	0.548	10	9	0.881
Positive	71	59	12	-	36	35	-
Serum AFP, ug/L	-	-	-	-	-	-	-
<400	57	46	11	0.896	31	26	0.414
≥400	33	27	6	-	15	18	-

HCC, hepatocellular carcinoma; AFP, alpha-fetoprotein; “-” represents the meaning of “No Refer”. * Fisher’s exact test indicates *p* < 0.05. Values in bold are statistically significant.

**Table 2 pharmaceutics-15-01887-t002:** Correlation of CPEB2 and miR-210-3p levels in HCC.

	No. of Cases with CPEB2^Low^	No. of Cases with CPEB2^High^	*p*-Value
No. of cases with miR-210-3p^Low^	33	13	-
No. of cases with miR-210-3p^High^	40	4	0.02
Fisher’s exact test	-	-	-

**Table 3 pharmaceutics-15-01887-t003:** Small molecules affecting miR-210 expression in Homo sapiens. The table summarizes the small molecules found to downregulate miR-210.

miRNA	Small Molecule	DrugbankID	CID	Detection Method	Species	PMID	Year	Expression
miR-210	Lenalidomide	DB00480	216326	Quantitative real-time PCR	Homo sapiens	25287904	2014	downregulated
miR-210	Arsenic trioxide	DB01169	14888	Microarray	Homo sapiens	17108120	2006	downregulated
miR-210	5-Fluorouracil	DB00544	3385	Quantitative real-time PCR	Homo sapiens	17702597	2007	downregulated
miR-210	Ginsenoside Rh2		119307	Microarray	Homo sapiens	21372826	2011	downregulated
miR-210	Aidi injection			Microarray	Homo sapiens	21563499	2011	downregulated
miR-210	5-aza-2′-deoxycytidine	DB01262	451668	Microarray	Homo sapiens	22076154	2011	downregulated
miR-210	5-aza-2′-deoxycytidine	DB01262	451668	Microarray	Homo sapiens	22076154	2011	downregulated
miR-210	5-aza-2′-deoxycytidine	DB01262	451668	Microarray	Homo sapiens	22076154	2011	downregulated
miR-210	Trastuzumab	DB00072		Microarray	Homo sapiens	22384020	2012	downregulated
miR-210	Trastuzumab	DB00072		Quantitative real-time PCR	Homo sapiens	22384020	2012	downregulated

## Data Availability

All data will be provided upon reasonable request.

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
