# Peer review of "CPEB2 Suppresses Hepatocellular Carcinoma Epithelial–Mesenchymal Transition and Metastasis through Regulating the HIF-1α/miR-210-3p/CPEB2 Axis"

_pharmaceutics, 2023, doi:10.3390/pharmaceutics15071887_

Round 1

Reviewer 1 Report

In the present work You et al. attempted to establish the role of CPEB2 and miR-210 in Hepatocellular Caricinoma (HCC). The authors have performed a great deal of experimentation in order to investigate their hypothesis and they have went in-depth for the mechanisms in question.

I have really enjoyed reading their manuscript, which I found very interesting. I would also say that the authors have attempted to implement a new idea on tumor therapeutics. I agree with their statement, that most transcriptome-related treatments are based on inhibiting gene expression of tumor-promoting genes, whereas the stimulation of tumor-inhibiting genes is still very challenging. The only way to achieve this is by targeting gene expression, indirectly, gene expression regulators (as for example miRs), which the authors succeeded to manifest. Besides some typos and small grammar errors, I don’t have any major comments for the present work.

I have two minor comments; I suggest to mention the limitations of the study including the limitations of enhancing, in vivo, tumor suppressing genes. In addition, I would like to see a comment/hypothesis, in the “Discussion” section, on the probable mechanisms small chemical molecules can have an effect on miR expression (in particular miR-210 if possible). What can be done better, or what are the next steps?

Finally, the authors should highlight their results and mention how their findings could prove useful for the treatment of HCC.

Author Response

 Dear Reviewer#1,

Thank you for your valuable suggestions. The following is my response to your comment.

  1. I suggest to mention the limitations of the study including the limitations of enhancing, in vivo, tumor suppressing genes.

Response: Thank you very much for your valuable advice. According to your advice, we have elaborated on the limitations of this study in the 4th paragraph (marked in red) of the “Discussion” section of the manuscript.

  1. In addition, I would like to see a comment/hypothesis, in the “Discussion” section, on the probable mechanisms small chemical molecules can have an effect on miR expression (in particular miR-210 if possible). What can be done better, or what are the next steps?

Response: Thank you very much to point out the issues in our manuscript. We have made detailed revisions and additions in the 2nd paragraph (marked in red) of the “Discussion” section based on your suggestions.

  1. Finally, the authors should highlight their results and mention how their findings could prove useful for the treatment of HCC.

Response: Thank you very much for your valuable suggestions. We have carefully revised the “Conclusion” section of the manuscript and highlighted our results, please see 1st paragraph (marked in red) of the “Discussion” section for an update.

Reviewer 2 Report

The Title of the manuscript implies that the authors have developed a novel strategy for drug detection.  This isn't actually precise as the only mention of drug interaction with miR-210-3p derives from the literature at the end of the manuscript in the form of Table 3.  There are no experiments in this manuscript to demonstrate these drugs effectively bind to miR-210-3p.  The strategy in this manuscript is the association between levels of CPEB and miR-210-3p and its affects on OS in HCC.  This correlation is important and the fact that drugs targeting miR-210-3p may also be available helps to imagine possible therapeutic benefits for HCC.

There appears to be some confusion in Fig. 5 and Section 3.6 as to the direction of the targeting of CPEB. CPEB does not target miR-210-3p (Fig. 5), rather Fig. 5 validates binding or non binding of miR-210-3p in wt versus mt 3'UTR sequences of CPEB.  In other words, miR-210-3p binds to the 3'UTR of CPEB and therefore, CPEB is the target of miR-210-3p.  In section 3.6, you do not intend to suggest that miR-210-3p has any influence beyond what you observe in CPEB siRNA experiments.  Section 3.6 describes the interaction and reliance on miR-210-3p for CPEB activity. A better way of saying what you observe is miR-210-3p's influence on HCC metastasis and EMT processes is mediated thru CPEB.    

The English grammar is acceptable, particularly in the results and conclusion sections.  The abstract needs some attention in line 15 "by and by" and the abbreviation for cytoplasmic polyadenylation element binding protein (line 18) is already used in line 15. In the introduction, lines 45-48 need some attention and on line 53, it's not vital "parts" but vital "roles".  

Reviewer 3 Report

The submitted manuscript entitled “A Novel Strategy for Hepatocellular Carcinoma Drug Prediction: Integrating Transcriptomics and Pharmacogenomics through CPEB2 and miR-210-3p” is focused on investigating of regulation of CPEB2 expression and its direct targeting by miR-210-3p and effects of drugs on CPEB2/miR-210 interaction in HCC. This study scientifically sounds and may be of interest for the journal audience. However, there are some concerns and recommendations to improve the quality of the manuscript. There are as follows:

1.     The title of the manuscript is somewhat confusing; it contains the phrase “A novel strategy”, however, as mentioned in the Abstract “CPEB2 expression by targeting miR-210-3p” is not a strategy, because strategy is a plan of actions. This should be changed.

2.     Only a little part of the study was focused on the investigation of drug response in HCC patients. i.e. elucidating drugs effecting CPEB2/miR-210 interaction. This also dictated necessity to change the title of the manuscript.

3.     The manuscript contains only 16 References and only 2 of them were published in the last 5 years. There are no references in the Discussion section.  It is recommended to add more references, especially, in the Discussion section to provide background for this study and comparative interpretations of the obtained results with other studies. For example, the authors should provide a background for usage of protein and miRNA biomarkers to predict drug response in HCC by citing the following papers: doi: 10.1080/14737159.2021.1987217 and doi: 10.3748/wjg.v25.i29.3870. To cmpare with other studies: doi: 10.1042/BJ20081353, doi: 10.2478/bjmg-2022-0007, doi: 10.1093/nar/gkq635, etc.

4.     English language style and grammar require editing. Section 3.2, rephase title because it is not clear that “CPEB2 inhibits HCC cell… “. Also,  on lines 201-202, “respectively” should be added; “is a key link to drives tumor cell metastasis” (line 47); “AFP, alpha fetal protein”- it should be alpha-fetoprotein (Table 1, line 99); “Lysed Cells on ice” (line 110); “Cancer cells (5×106)” (line 159); on line 335, change “have been” for “were” because these results were obtained in this study, etc.

English laguage requires editing
